# Pharmacological Modulation of the MIP-1 Family and Their Receptors Reduces Neuropathic Pain Symptoms and Influences Morphine Analgesia: Evidence from a Mouse Model

**DOI:** 10.3390/brainsci13040579

**Published:** 2023-03-29

**Authors:** Agata Ciechanowska, Katarzyna Pawlik, Katarzyna Ciapała, Joanna Mika

**Affiliations:** Department of Pain Pharmacology, Maj Institute of Pharmacology Polish Academy of Sciences, 12 Smetna Str., 31-343 Kraków, Poland; ciechan@if-pan.krakow.pl (A.C.); pawlik@if-pan.krakow.pl (K.P.); kat.ciapala@gmail.com (K.C.)

**Keywords:** CCL3, CCL4, CCL9, CCR1 antagonist (J113863), CCR5 antagonist (TAK-220, AZD-5672)

## Abstract

Neuropathic pain pathophysiology is not fully understood, but it was recently shown that MIP-1 family members (CCL3, CCL4, and CCL9) have strong pronociceptive properties. Our goal was to examine how pharmacological modulation of these chemokines and their receptors (CCR1 and CCR5) influence hypersensitivity after nerve injury in Albino Swiss male mice. The spinal changes in the mRNA/protein levels of the abovementioned chemokines and their receptors were measured using RT-qPCR and ELISA/Western blot techniques in a mouse model of chronic constriction injury of the sciatic nerve. Behavioral studies were performed using the von Frey and cold plate tests after pharmacological treatment with neutralizing antibodies (nAbs) against chemokines or antagonists (CCR1-J113863, CCR5-TAK-220/AZD-5672) alone and in coadministration with morphine on Day 7, when the hypersensitivity was fully developed. Our results showed enhanced protein levels of CCL3 and CCL9 1 and 7 days after nerve injury. The single intrathecal administration of CCL3 or CCL9 nAb, J113863, TAK-220, or AZD-5672 diminished neuropathic pain symptoms and enhanced morphine analgesia. These findings highlight the important roles of CCL3 and CCL9 in neuropathic pain and additionally indicate that these chemokines play essential roles in opioid analgesia. The obtained results suggest CCR1 and CCR5 as new, interesting targets in neuropathy treatment.

## 1. Introduction

Neuropathy affects up to 10% of humans [1] and is induced by varied causes, including direct damage of the nerves. Even the most effective painkillers in general (opioid drugs) are often ineffective against painful neuropathy [2]. This issue requires further studies explaining neuro-immunological processes in order to improve the treatments [3]. Neuropathic pain states are distinguished by activation and/or an influx of immune and glial cells in the spinal cord. Within the central nervous system (CNS), three main types of glial cells are present under physiological conditions: oligodendrocytes, astrocytes, and microglia (being the least abundant) [4]. Microglial cells along with macrophages have an indisputable role in the preservation of CNS homeostasis and can take on many activation states leading to changes in morphology, gene expression and function [5]. Additionally, astroglia are crucial in maintaining the balanced functions of the CNS. Their activation has neurodegenerative properties, inter alia, in the case of neuropathic pain [6]. Recent studies have suggested that oligodendrocytes are activated in neuropathy and that they participate in nociceptive transmission [7]. After being activated, the abovementioned cells can release multiple factors, such as chemokines, which are especially important for nociception [3]. Chemokines have low molecular weight but strong chemotactic properties [8]. However, their roles go far beyond chemotaxis. Previous studies have suggested that the blockade of chemokines from different families (e.g., CCL1, CXCL3, or XCL1) by neutralizing antibodies provides analgesic effects [9,10,11,12]. Likewise, the blockade of chemokine receptors was shown to be a promising therapeutic strategy in mouse models of neuropathic pain. For example, J113863 (CCR1), RS504393 (CCR2), C021 (CCR4), maraviroc (CCR5), and NVPCXCR2 20 (CXCR2) reduced hypersensitivity and enhanced the effectiveness of morphine [10,13,14,15].

In our previous studies, we showed that chemokines from the MIP-1 (macrophage inflammatory protein-1) family, including CCL3 (MIP-1-alpha), CCL4 (MIP-1-beta), and CCL9 (MIP-1-gamma), have strong pronociceptive properties [13,16] and are engaged in the development of neuropathic pain caused by diabetes [13], which is a metabolic polyneuropathy model. It has also been shown by some studies in mice and rats that CCL3 and CCL9 are upregulated in the spinal cord and/or DRG levels after mechanical damage to a single nerve [14,17,18]. However, the detailed participation of MIP-1 family ligands and receptors after chronic constriction injury of the sciatic nerve (CCI model) has not been studied in depth. This fact convinced us that there is still a lack of precise knowledge about the specific activity and function of this family of chemokines. Therefore, we decided to check, at the mRNA and protein levels, if there were any changes in the amount of chemokines from the MIP-1 family and their receptors (CCR1: CCL3, CCL4, CCL9, and CCR5: CCL3 and CCL4) in the lumbar spinal cord of mice after CCI. In addition, we simultaneously verified the changes in cell markers for oligodendrocytes, microglia, microglia/macrophages, and astroglia. Then, via the administration of CCL3 and CCL9 nAbs, we investigated the influence of these two chemokines on neuropathic pain symptoms (mechanical and thermal hypersensitivity). In the next experiment, we studied whether these specific antibodies coadministered with morphine acted better than morphine alone. Finally, we investigated whether a blockade of receptors of the MIP-1 family chemokines by J113863 (CCR1 antagonist), TAK-220 (CCR5 antagonist), or AZD-5672 (CCR5 antagonist) had any ability to relieve neuropathic pain symptoms that developed after CCI in comparison to simultaneous CCR1 and CCR5 blockade via the coadministration of J113863 with TAK-220 or AZD-5672. Additionally, we measured whether these antagonists coadministered with morphine had better analgesic potential than morphine alone.

## 2. Materials and Methods

### 2.1. Animals

Our experiments were performed on mice (Albino Swiss strain) from Charles River (Germany). We used adult male mice (9–11 weeks old, weighing 22–27 g). They were kept in the following conditions: 22 ± 2 °C, 55 ± 10% relative humidity, 12/12 h light/dark cycle. Animals had limitless access to nourishment and were kept in cages with an enriched environment (blocks of wood). The Ethics Committee of the Maj Institute of Pharmacology of the Polish Academy of Sciences supervised and permitted the procedures (LKE: permission numbers: 1277/2015, 301/2017, 75/2017, 305/2017, 235/2020, and 40/2023). According to the 3R policy, we used the minimal, essential number of animals. In the graphs with the results of behavioral tests, we have also included values for naive animals. In order to control the level of neuropathic pain symptoms in mice on Day 7, we performed calculations showing that the naive group differed significantly from animals after CCI at all time points examined, in both behavioral tests. The differences between the naive and vehicle-treated CCI-exposed mice and in all studied time points were significant in the von Frey test (*p* < 0.0001) and in the cold plate test (*p* < 0.0001) on Day 7, when mechanical and thermal hypersensitivity were fully developed.

### 2.2. Chronic Constriction Injury

According to the procedure by Bennet and Xie [19], chronic constriction injury (CCI) of the sciatic nerve was performed as follows. Mice were anesthetized via the inhalation of isoflurane (induction and maintenance 3%). Next, an incision was made below the right hip bone. After exposure, the right sciatic nerve was loosely tied three times and hidden. In the next step, the skin was sutured, and animals were left to rest and regenerate. All operated-on mice developed neuropathic-pain-related behaviors. Behavioral tests were performed 1, 4, 7, 14, and 35 days after the CCI procedure.

### 2.3. Biochemical Tests

The methods for mRNA and protein level measurements were those used in our lab for years [9,12,20].

#### 2.3.1. Analysis of Gene Expression via RT-qPCR

Immediately after decapitation, the spinal cords were removed from healthy (naive) and CCI-exposed mice on Days 1, 4, 7, 14, and 35. In the next step, the tissue was sectioned, and the lumbar (L4–L6) region of the spinal cord was isolated and placed into Eppendorf tubes^®^ with RNAlater (Ambion Inc., Austin, TX, USA) and frozen at −80 °C. Total RNA was extracted according to Chomczynski and Sacchi [21] with TRIzol reagent (Invitrogen, Carlsbad, CA, USA). Next, the concentration of mRNA was measured using a spectrophotometer (DeNovix DS-11, DeNovix Inc., Wilmington, SA, USA). Reverse transcription was performed with 1000 ng of total RNA by applying Omniscript Reverse Transcriptase (Qiagen Inc., Hilden, Germany) at 37 °C for 60 min. In the next step, the obtained cDNA was diluted with H_2_O at a proportion of 1:10. Finally, the RT-qPCR was performed with the use of Assay-On-Demand TaqMan probes in agreement with the manufacturer’s protocol (Applied Biosystems, Foster City, CA, USA) in an iCycler device (Bio-Rad, Hercules, Warsaw, Poland). The following TaqMan primers were used for this study: Mm00446968_m1 (Hprt), Mm01210556_m1 (Olig2), Mm00525305_m1 (TMEM119), Mm00479862_g1 (IBA1), Mm01253033_m1 (GFAP), Mm00441259_g1 (CCL3), Mm00443111_m1 (CCL4), Mm00441260_m1 (CCL9), Mm00438260_s1 (CCR1), and Mm01963251_s1 (CCR5). The housekeeping gene (the Hprt transcript) was quantified to control for variations in the amount of cDNA. The cycle threshold values were automatically calculated using the iCycler IQ 3.0 software with the default parameters. The abundance of RNA was calculated as 2^−(threshold cycle)^.

#### 2.3.2. Enzyme-Linked Immunosorbent Assay (ELISA) Analysis

Immediately after decapitation, the spinal cords were removed from healthy (naive) and CCI-exposed mice on Days 1, 7, and 35. In the next step, the tissue was sectioned, and the lumbar (L4–L6) region of the spinal cord was isolated and placed into Eppendorf tubes^®^ with RNAlater (Ambion Inc., Austin, TX, USA) and used for ELISAs according to the manufacturer’s recommendations. The tissue homogenates were fixed in RIPA buffer with a protease and phosphatase inhibitor cocktail (Sigma-Aldrich, St. Louis, MI, USA). Next, the samples were centrifuged (14,000 rpm, for 30 min at 4 °C). The concentration of total protein was measured using the bicinchoninic acid method. The level of protein was measured in the tissue homogenates using Mouse CCL3/MIP-1-Alpha, Sandwich ELISA, LS-F4952, LifeSpan Biosciences, Seattle, WA, USA; Mouse CCL4/MIP-1 Beta, Sandwich ELISA, LS-F4954, LifeSpan Biosciences; and Mouse CCL9/MIP-1 Gamma, Sandwich ELISA, LS-F55161, LifeSpan Biosciences. The detection ranges were as follows: CCL3: 15.6–1000 pg/mL; CCL4: 62.5–4000 pg/mL; and CCL9: 7.8–500 pg/mL. The manufacturer provided positive controls for each assay.

#### 2.3.3. Western Blot Analysis

Immediately after decapitation, the spinal cords were removed from healthy (naive) and CCI-exposed mice on Days 1, 7, and 35. The tissue was sectioned, and the lumbar (L4–L6) region of the spinal cord was isolated and placed into Eppendorf tubes^®^ with RIPA buffer supplemented with a protease inhibitor cocktail (Sigma-Aldrich, St. Louis, MI, USA). Next, the samples were centrifuged (14,000 rpm, for 30 min at 4 °C). The concentration of total protein was measured using the bicinchoninic acid method. Samples of 10 μg protein were heated at 98 °C for 8 min in loading buffer (4 × Laemmli Buffer, Bio-Rad, Warsaw, Poland). The samples were loaded into precast polyacrylamide gels (4–15% Criterion TGX, Bio-Rad) and transferred to Immune-Blot PVDF membranes (Bio-Rad) through a semidry transfer system (30 min, 25 V). The membranes were blocked for 1 h with dry milk (5%, nonfat, Bio-Rad) reconstituted in Tris-buffered saline with Tween 20 (0.1% in TBST). The membranes were washed with TBST (2 min, 3 × 5 min) and incubated overnight at 4 °C with the following commercially available primary antibodies: mouse anti-β-actin (1:1000; Merck, Darmstadt, Germany), rabbit anti-IBA1 (1:500, Novus Biologicals; Centennial, CO, USA), rabbit anti-GFAP (1:10,000, Novus Biologicals, Centennial, CO, USA), rabbit anti-CCR1 (1:500; Novus Biologicals, Centennial, CO, USA), and rabbit anti-CCR5 (1:500; Novus Biologicals, Centennial, CO, USA). Then, the membranes were incubated in horseradish-peroxidase-conjugated anti-rabbit or anti-mouse secondary antibodies for 1 h at room temperature (Vector Laboratories, Burlingame, CA, USA) at a dilution of 1:5000. The SignalBoost Immunoreaction Enhancer Kit (Merck, Darmstadt, Germany) was used as a dissolvent for the primary and secondary antibodies. The membranes were washed again in TBST (2 min, 3 × 5 min). Clarity Western ECL Substrate (Bio-Rad, Warsaw, Poland) was used to detect the immune complexes, and they were visualized using the Fujifilm LAS-4000 Fluor Imager system. Fujifilm Multi Gauge was used for the quantification of relative levels of the immunoreactive bands.

### 2.4. Single Intrathecal Drug Administration in the Mouse Model of Neuropathy

Substances were administered through lumbar puncture intrathecally (i.t.) in a volume of 5 μL between the L5 and L6 vertebrae (Hylden and Wilcox 1980, with modifications by Fairbanks 2003) [18,19]. For injection, a Hamilton syringe with a thin needle (0.3 × 13 mm) was used. Half an hour before the administration of each pharmacological tool, we performed the pretest which is the first measurement of each animal during the course of an experiment. It is intended to show whether the animals used in the particular experiment have developed symptoms of hypersensitivity to mechanical and thermal stimuli properly.

This is a standard procedure in our laboratory [9,12,20].

#### 2.4.1. Administration of CCL3 and CCL9 Neutralizing Antibodies

A single i.t. administration of CCL3 nAb (AF-450-NA, Mouse CCL3/MIP-1 alpha Antibody, R&D Systems; Minneapolis, MI, USA) or CCL9 nAb (AF463, Mouse CCL9/10/MIP-1 gamma Antibody, R&D Systems) was administered to CCI mice at the dose of 0.5, 2, or 4 µg/5 µL on Day 7, when mechanical and thermal hypersensitivity were fully developed. Behavioral testing was performed after 1, 4, and 24 h. CCL3 and CCL9 nAbs were dissolved in PBS (Merck; Darmstadt, Germany), and PBS was injected as the vehicle (V) in the control group. The details about using neutralizing antibodies are available on the manufacturer’s pages: CCL3 nAb [22] and CCL9 nAb [23].

#### 2.4.2. Administration of CCL3 Neutralizing Antibody with Morphine

The i.t. administration of CCL3 neutralizing antibody (2 μg/5 μL) followed by the i.t. administration of morphine (TEVA; Krakow; Poland; 2.5 μg/5 μL) was performed 7 days after CCI. The dose of nAb was chosen based on the results obtained from the above experiment. The doses of opioid were chosen based on our previous study [24]. First, animals received V or CCL3 nAb. Next, after 3 h, there was a second injection of *aqua pro injectione* (W) or morphine (M). The von Frey and cold plate tests were performed 0.5 h after the second administration (W or M), which was 3.5 h after the first administration (V or CCL3 nAb)

#### 2.4.3. Administration of CCL9 Neutralizing Antibody with Morphine

The i.t. administration of CCL9 neutralizing antibody (2 μg/5 μL) followed by the i.t. administration of morphine was performed 7 days after CCI. The dose of nAb was chosen based on the results obtained above. First, animals received V or CCL9 nAb. After 1 h, there was a second injection of W or M. The von Frey and cold plate tests were performed 0.5 h after the second administration (W or M), which was 1.5 h after the first administration (V or CCL9 nAb).

#### 2.4.4. Administration of CCR1 and CCR5 Antagonists

A single i.t. administration of J113863 (CCR1 antagonist, at doses of 1, 15, 30, and 60 µg/5; µL cat. #2595, Tocris, Bristol, UK), TAK-220 (CCR5 antagonist, at doses of 0.5, 2, 4, and 15 µg/5; µL cat. #HY-19974/CS-5579, MedChemExpress, Monmouth Junction, NJ, USA), or AZD-5672 (CCR5 antagonist, at doses of 0.5, 2, 4, and 15 µg/5 µL cat. #HY-119101/CS-0068004, MedChemExpress) were administered once to CCI-exposed mice on Day 7, when mechanical and thermal hypersensitivity were fully developed. Behavioral testing was performed after 1, 4, and 24 h. CCR1 and CCR5 antagonists were dissolved in dimethyl sulfoxide (DMSO, cat. #D8418, Sigma-Aldrich, Saint Louis, MI, USA), and DMSO was used as the vehicle (V). The details about the agonists used are available on the manufacturer’s pages: J113863 [25], TAK-220 [26], and AZD-5672 [27].

#### 2.4.5. Administration of CCR1 Antagonist with Morphine

The i.t. administration of J113863 (15 μg/5 μL) followed by the i.t. administration of morphine was performed 7 days after CCI. The dose of the antagonist was based on the obtained results. First, animals received V or J113863. Next, after 3 h, there was a second injection of W or M. The von Frey and cold plate tests were performed 0.5 h after the second administration (W or M) and 3.5 h after the first administration (V or J113863).

#### 2.4.6. Administration of CCR5 Antagonists with Morphine

The i.t. administration of TAK-220 or AZD-5672 followed by the i.t. administration of morphine was performed 7 days after CCI. The doses of antagonists were based on the obtained results. First, animals received V, TAK-220, or AZD-5672. After 1 h, there was a second injection of W or M. The von Frey and cold plate tests were performed 0.5 h after the second administration (W or M) and 1.5 h after the first administration (V, TAK-220, or AZD-5672).

#### 2.4.7. Coadministration of CCR1 and CCR5 Antagonists

The i.t. administration of J113863 (15 μg/5 μL) with the i.t. administration of TAK-220 or AZD-5672 (15 μg/5 μL) was performed 7 days after CCI, and the doses were based on the above experimental results. Animals received V, antagonist of CCR1 (J113863), antagonist of CCR5 (TAK-220 or AZD-5672), or their combination (J113863 + TAK-220 or J11 + AZD-5672). Behavioral testing was performed after 1 and 4 h.

### 2.5. Behavioral Tests

#### 2.5.1. Von Frey Test

For the measurement of tactile hypersensitivity, we used calibrated nylon monofilaments (ranging from 0.6–6 g, Stoelting, Wood Dale, IL, USA). First, the mice were put into plastic cages with a floor of wire mesh. After 5 min of adaptation, the reactions to mechanical stimuli were checked by the application of von Frey filaments. The measurement started with the thinnest filament (0.6 g) and continued until the hind paw was lifted. The pressure of the filament that caused the reaction was recorded as the result. If this reaction was not observed, filaments were applied to increase pressure (g) until the last filament used had a pressing force of 6 g, which was the cutoff latency [10]. The measurement with all filaments used was always based on three touches, which followed each other directly, of the midplantar surface of the hind paw. Each mouse (naive and CCI-exposed individuals) was measured once in every time point. Only the injured paw (right paw in naive) was measured and the result of the test was the value of the filament that caused the reaction of the tested mouse. These responses included paw withdrawal and shaking. If the outcome of the test was unclear, we repeated the measurements of individual mice after 5 min. This is a standard test used in our laboratory [9,12,20].

#### 2.5.2. Cold Plate Test

For the measurement of thermal hypersensitivity, we used a cold plate/hot plate analgesia meter (Ugo Basile, Gemonio, Italy). The temperature of the plate surface was kept at 2 °C. The animals were placed on the chilled surface of the plate and observed until they lifted their hind paw, and the time of the reaction was recorded as described previously [10]. The maximal possible time for the animals to stand on the plate was 30 s, which was the cutoff latency. After CCI, the foot with the constricted nerve was always the first to react. This is a standard test used in our laboratory [9,12,20].

### 2.6. Statistical Analysis

The data obtained in the behavioral experiments (von Frey, cold plate tests) are presented as mean ± SEM. The data obtained in biochemical experiments (RT-qPCR, Western blot, and ELISA analyses) are presented as fold change relative to the control group (naive mice) ± SEM. The obtained results were statistically evaluated using one-way ANOVA with Bonferroni’s post hoc test for multiple comparisons. Some of the results were evaluated using two-way ANOVA to detect time × drug interaction. All of the statistical analyses were performed using Prism (ver. 8.1.1 (330), GraphPad Software, Inc., San Diego, CA, USA).

## 3. Results

### 3.1. Temporal Changes in the mRNA and/or Protein Levels of Olig2, TMEM119, IBA1, and GFAP Measured in Parallel with Pain-Related Behavior after Chronic Constriction Injury of the Sciatic Nerve in Mice

Chronic constriction injury of the sciatic nerve led to the development of thermal hypersensitivity between Days 1 (*p* < 0.0001) and 35 (*p* < 0.0001) (Figure 1B). Over this time, we measured the mRNA and/or protein expression changes in cell markers (Figure 1D).

RT-qPCR analysis showed that the mRNA level of a marker for oligodendrocytes (*Olig2*) was not changed after CCI (Figure 1A). However, the mRNA level of the microglial marker (*TMEM119)* was strongly elevated between the 4th (*p* = 0.0011) and 7th days (*p* = 0.0014) after injury of the nerve (Figure 1C); similarly, the microglia/macrophage marker *IBA1* was elevated at the mRNA level from the 1st (*p* = 0.0442) to the 14th day (*p* = 0.0003) (Figure 1E). The astroglial marker (*GFAP)* was also elevated at the mRNA level, but only on the 4th (*p* = 0.0241) and 14th days (*p* = 0.0351) (Figure 1G).

Western blot analysis showed that the protein level of a microglia/macrophage marker (IBA1) was elevated on Day 7 after CCI (*p* = 0.0005) (Figure 1F). The protein level of the astroglial marker (GFAP) was elevated on the 7th (*p* = 0.0367) and 35th days (*p* = 0.0087) (Figure 1H).

### 3.2. Temporal Changes in the mRNA and Protein Levels of CCL3, CCL4, and CCL9 after Chronic Constriction Injury of the Sciatic Nerve in Mice

Chronic constriction injury evoked changes in the mRNA level of CCL3, which was upregulated between the 4th (*p* = 0.0246) and 35th days (*p* = 0.0006) (Figure 2A). CCL4 was elevated from the 4th (*p* = 0.0043) to the 35th day (*p* = 0.0105) (Figure 2C), and similarly the level of CCL9 grew significantly from the 4th (*p* < 0.0001) to the 35th day (*p* = 0.0006) (Figure 2E).

Changes in the protein level were significant in the case of CCL3 between the 1st (*p* = 0.0054) and 7th days (*p* = 0.0307) (Figure 2B), and likewise for CCL9 (*p* = 0.0019 and *p* = 0.0295, respectively) (Figure 2F). A slight decrease in the CCL4 level was observed 35 days after CCI (*p* = 0.0346).

### 3.3. Effects of a Single Intrathecal Administration of CCL3 nAb on Pain-Related Behavior and Morphine Analgesia 7 Days after Chronic Constriction Injury of the Sciatic Nerve in Mice

CCL3 nAb was administered at doses of 0.5, 2, and 4 μg/5 μL (Figure 3A). In the von Frey test, a significant reduction in the mechanical hypersensitivity was observed 1 h after the two higher doses of 2 μg/5 μL (*p* = 0.0018) and 4 μg/5 μL (*p* < 0.0001), and 4 h after the administration of 2 μg/5 μL (*p* < 0.0001) (Figure 3B). In the cold plate test, there was also a significant reduction in the thermal hypersensitivity observed after 1 h, but only after the dose of 4 μg/5 μL (*p* = 0.0205). At 4 h, an effect was observed after the two higher doses of 2 μg/5 μL (*p* = 0.0002) and 4 μg/5 μL (*p* < 0.0001) (Figure 3C). Two-way ANOVA confirmed a significant interaction between the treatment and the analyzed time point (von Frey: *p* < 0.0001; cold plate: *p* < 0.0001).

Additionally, we measured the influence of CCL3 nAb on analgesia evoked by morphine at a dose of 2.5 μg/5 μL (Figure 3D). Morphine alone significantly reduced thermal hypersensitivity (*p* < 0.0001) (Figure 3F). The observed outcome of CCL3 nAb with morphine coadministration strongly reduced both mechanical (Figure 3E) and thermal (Figure 3F) hypersensitivity compared to morphine and CCL3 nAb administered alone.

### 3.4. Effects of a Single Intrathecal Administration of CCL9 nAb on Pain-Related Behavior and Morphine Analgesia 7 Days after Chronic Constriction Injury of the Sciatic Nerve in Mice

CCL9 nAb was administered at doses of 0.5, 2, and 4 μg/5 μL (Figure 4A). The mechanical threshold measured using the von Frey test was significantly reduced observed after 1 h for the two higher doses of 2 μg/5 μL (*p* = 0.0002) and 4 μg/5 μL (*p* = 0.0141), but 4 h after administration only for the dose of 4 μg/5 μL (*p* = 0.0026) (Figure 4B). At the same time, in the case of the thermal threshold (cold plate test), a significant reduction was observed 1 h after the dose of 2 μg/5 μL (*p* = 0.0036) and 4 h after the two higher doses of 2 μg/5 μL (*p* = 0.0019) and 4 μg/5 μL (*p* = 0.0371) (Figure 4C). Two-way ANOVA confirmed a significant interaction between the treatment and the analyzed time point (von Frey: *p* < 0.0001; cold plate: *p* = 0.0089).

Furthermore, we measured the influence of CCL9 nAb on morphine analgesia (Figure 4D). Morphine administration alone significantly reduced mechanical (*p* = 0.0466) and thermal (*p* = 0.0052) (Figure 4E,F) hypersensitivity. However, the CCL9 nAb and morphine coadministration strongly reduced both mechanical (Figure 4E) and thermal (Figure 4F) hypersensitivity and was more effective than morphine and/or CCL9 nAb administered alone.

### 3.5. Temporal Changes in the mRNA and Protein Levels of CCR1 and CCR5 after Chronic Constriction Injury of the Sciatic Nerve in Mice

Chronic constriction injury evoked changes in the mRNA level of *CCR1*, which increased between the 4th (*p* = 0.0251) and 35th days (*p* = 0.0065) (Figure 5A). For the mRNA level of *CCR5*, we also observed a significant increase, but only between the 4th (*p* = 0.0032) and 14th days (*p* = 0.0208) (Figure 5C).

At the protein level, there were no changes in CCR5 (Figure 5D). There was a slight decrease in the protein level of CCR1 7 days after CCI (*p* = 0.0182) (Figure 5B).

### 3.6. Effects of a Single Intrathecal J113863 Administration on Pain-Related Behavior and Morphine Analgesia 7 Days after Chronic Constriction Injury of the Sciatic Nerve in Mice

J113863 was administered at doses of 1, 15, 30, and 60 μg/5 μL (Figure 6A). In the von Frey test, a significant reduction in mechanical hypersensitivity was observed 1 h after the doses of 1 μg/5 μL (*p* = 0.0400), 15 μg/5 μL (*p* = 0.0041), 30 μg/5 μL (*p* < 0.0001), and 60 μg/5 μL (*p* = 0.0056). J113863 was even more effective 4 h after administration at doses of 15 μg/5 μL (*p* = 0.0004) and 60 μg/5 μL (*p* < 0.0001) (Figure 6B). In the cold plate test, the highest reduction in thermal hypersensitivity was observed 1 h after doses of 1 μg/5 μL (*p* = 0.0038), 15 μg/5 μL (*p* = 0.0014), 30 μg/5 μL (*p* = 0.0023), and 60 μg/5 μL (*p* < 0.0001). Similarly, a reduction in thermal hypersensitivity was observed 4 h after doses of 1 μg/5 μL (*p* = 0.0192), 15 μg/5 μL (*p* = 0.0031), 30 μg/5 μL (*p* = 0.0022), and 60 μg/5 μL (*p* < 0.0001) (Figure 6C). Two-way ANOVA found a significant interaction between treatment and time (von Frey: *p* < 0.0001; cold plate: *p* < 0.0001).

Subsequently, we measured the influence of J113863 on morphine analgesia (Figure 6D). Morphine alone (2.5 μg/5 μL) significantly reduced mechanical (*p* = 0.0347) and thermal (*p* < 0.0001) (Figure 6E,F) hypersensitivity. Even so, the effect of J113863 and morphine coadministration significantly reduced both mechanical (Figure 6E) and thermal (Figure 6F) hypersensitivity and was more potent than morphine and J113863 administered alone.

### 3.7. Effects of a Single Intrathecal TAK-220 Administration on Pain-Related Behavior and Morphine Analgesia 7 Days after Chronic Constriction Injury of the Sciatic Nerve in Mice

First, the CCR5 antagonist TAK-220 was administered at doses of 0.5, 2, 4, and 15 μg/5 μL (Figure 7A). In the test for mechanical hypersensitivity after 1 h, the significant reduction in pain symptoms was induced by the doses of 2 μg/5 μL (*p* = 0.0054), 4 μg/5 μL (*p* < 0.0001), and 15 μg/5 μL (*p* < 0.0001). After 4 h, there was also a significant effect of the same doses of 2 μg/5 μL (*p* = 0.0358), 4 μg/5 μL (*p* = 0.0067), and 15 μg/5 μL (*p* < 0.0001) (Figure 7B). In the test for thermal hypersensitivity, there was a similar pattern of efficacy, and the most potent reduction was observed 1 h after doses of 2 μg/5 μL (*p* = 0.0042), 4 μg/5 μL (*p* = 0.0004), and 15 μg/5 μL (*p* < 0.0001). The effect was also significant 4 h after doses of 2 μg/5 μL (*p* = 0.0014) and 4 μg/5 μL (*p* = 0.0026) (Figure 7C). Two-way ANOVA found a significant interaction between treatment and time (von Frey: *p* < 0.0001; cold plate: *p* < 0.0001).

In the next step, we measured the influence of TAK-220 on morphine analgesia (Figure 7D). Morphine alone (2.5 μg/5 μL) significantly reduced mechanical (*p* = 0.0209) and thermal (*p* = 0.0018) (Figure 7E,F) hypersensitivity. Even so, the potentiation of morphine analgesia by the use of TAK-220 was substantial in the test of mechanical (*p* = 0.0093) (Figure 7E) hypersensitivity. Nevertheless, the effect of TAK-220 and morphine coadministration reduced both mechanical (Figure 7E) and thermal (Figure 7F) hypersensitivity and was more effective than morphine and/or TAK-220 administered alone

### 3.8. Effects of a Single Intrathecal AZD-5672 Administration on Pain-Related Behavior and Morphine Analgesia 7 Days after Chronic Constriction Injury of the Sciatic Nerve in Mice

The second CCR5 antagonist chosen for this study, AZD-5672, was also administered at doses of 0.5, 2, 4, and 15 μg/5 μL (Figure 8A). In the von Frey test (mechanical threshold), the significant reduction was observed 1 h after doses of 0.5 μg/5 μL (*p* = 0.0007), 2 μg/5 μL (*p* = 0.0010), and 4 μg/5 μL (*p* = 0.0140), and 4 h after doses of 0.5 μg/5 μL (*p* = 0.0001) and 15 μg/5 μL (*p* = 0.0488) (Figure 8B). In the cold plate test (thermal threshold), there was also a significant reduction observed 1 h after doses of 2 μg/5 μL (*p* = 0.0041), 4 μg/5 μL (*p* = 0.0024), and 15 μg/5 μL (*p* = 0.0017), and 4 h after doses of 4 μg/5 μL (*p* = 0.0010) and 15 μg/5 μL (*p* = 0.0426) (Figure 8C). Two-way ANOVA found a significant interaction between treatment and time (von Frey: *p* = 0.0006; cold plate: *p* = 0.0445).

Afterward, we measured the influence of AZD-5672 on morphine analgesia (Figure 8D). Compared to morphine administered alone (2.5 μg/5 μL), which significantly lowered mechanical (*p* = 0.0142) and thermal (*p* < 0.0001) (Figure 8E,F) hypersensitivity, coadministration with AZD-5672 was better than morphine and AZD-5672 administered alone but only for reducing mechanical (Figure 8E) hypersensitivity.

### 3.9. Comparison of the Effects of Intrathecal Administration of Substances Targeting CCR1 (J113863), CCR5 (TAK-220/AZD-5672), and Their Combination (J11 + TAK-220 or J11 + AZD-5672) on Pain-Related Behavior 7 Days after Chronic Constriction Injury of the Sciatic Nerve in Mice

We next measured the influence of a single i.t. coadministrations of CCR1 (J113863) and CCR5 (TAK-220/AZD-5672) antagonists at 1 and 4 h (Figure 9A); the times were selected according to the above results. After 1 h, there was greater analgesia by TAK-220 (Figure 7) and AZD-5672 (Figure 8), but after 4 h, there was a higher analgesia by J113863 (Figure 6) (at least in one behavioral test).

After 1 h, in the von Frey test, when the substances were administered alone, there was a far weaker analgesic effect observed for J113863 (*p* < 0.0001) and AZD-5672 (*p* = 0.0004) than for TAK-220 (Figure 9B). Moreover, the coadministration of J113863 + TAK-220 (*p* = 0.0168) and J113863 + AZD-5672 (*p* = 0.0066) was more effective than J113863 alone. The injections of J113863 + AZD-5672 were also more effective than single AZD-5672 (*p* = 0.0196) (Figure 9B). However, in the case of the cold plate test, there were no differences between groups, with the exception that the double blockade by J113863 + TAK-220 was slightly more successful than that by J113863 + AZD-5672 (*p* = 0.0440) (Figure 9C).

After 4 h, in the von Frey test, AZD-5672 was less analgesic than J113863 (*p* = 0.0465) and TAK-220 (*p* = 0.0251) (Figure 9D). However, in the cold plate test, J113863 alone was more effective than TAK-220 (*p* = 0.0082), J113863 + TAK-220 (*p* = 0.0062), and J113863 + AZD-5672 (*p* = 0.0039). There was no improvement in analgesia in groups receiving coadministration compared to single administration in the von Frey or cold plate test (Figure 9D,E).

## 4. Discussion

Our results indicated that in the CCI-induced neuropathic pain model, strong thermal hypersensitivity developed in parallel with the activation of macrophages, microglia, and astroglia, and in parallel we also observed enhanced protein levels of CCL3 and CCL9. These results correlate well with those obtained in behavioral studies, in which we showed for the first time that the administration of neutralizing antibodies for CCL3 and CCL9 showed analgesic effects on Day 7 in CCI-evoked neuropathy. Importantly, we obtained similar analgesic properties after a single intrathecal administration of J113863 (CCR1 antagonist), TAK-220, or AZD-5672 (CCR5 antagonists). Furthermore, blocking CCL3 or CCL9 and CCR1 or CCR5 led to the augmentation of the effectiveness of morphine, but CCR5 antagonists were only effective against mechanical hypersensitivity. Surprisingly, the coadministration of J113863 with TAK-220 or AZD-5672 was in general not far more effective against symptoms of neuropathic pain than either one alone. Our research emphasizes the important function of CCL3 and CCL9 and their receptors in the pathology of neuropathy and suggest their crucial role in opioid analgesia (Figure 1).

Neuropathic pain resulting from nerve injury is a highly impairing type of pain, which is often resistant to available treatments [28]. Chemokines have indisputable homeostatic functions based on attracting target cells to the place of their secretion, and what is especially important is that they can be produced not only by glial and immune cells, as originally thought, but also by neurons [29,30]. Data from recent years have confirmed that chemokines of the CC family, such as CCL2/3/4/5/7/8/9, have prominent pronociceptive properties after their intrathecal administration to naive mice [13,16]. We were interested in three of the abovementioned chemokines with strong algesic potential belonging to the MIP-1 family: CCL3, CCL4, and CCL9. In the lumbar spinal cord of mice with CCI-induced neuropathic pain, only two of them were highly upregulated at the protein level: CCL3 and CCL9. There was no upregulation in the protein level of CCL4, which conforms with the statements of Rojewska et al. [13] in a model of diabetic neuropathy. The results of our study indicate that changes in mRNA and protein levels of chemokines differ significantly, which is consistent with the latest literature [31]. It is well known that genetic information is converted from DNA to mRNA and then to proteins, but this does not necessarily involve translation. Determining the protein level of chemokines is very difficult because of their low molecular weight, and it has been attempted in very few studies. However, it is important, as our research shows, because it allows us to draw more accurate conclusions. Moreover, nerve injury results in a disruption of the blood–spinal cord barrier allowing for the time-dependent influx of peripheral immune cells [32]. Our research results show that among the MIP-1 family members, CCL3 and CCL9 play important roles in nociceptive pain transmission in neuropathy. Data from immunohistochemical analysis showed that CCL3 can be released by both neurons [13] and microglia [33]. Microglia were shown to produce CCL3 in primary cell cultures after ATP stimulation [34]. Therefore, we assume that shortly after CCI, neuronal cells secrete CCL3, which activates and attracts macrophages/microglia [13,34,35]. Later, microglia can also produce this chemokine, which can possibly act both in an autocrine and paracrine manner. However, this hypothesis needs further research. Our results demonstrated that neutralizing antibodies against CCL3 not only raised the nociceptive threshold but also enhanced the potency of morphine. Similar results were obtained in a model of diabetic neuropathy [13]. In our opinion, an understanding of the CCL3 role seems to be highly important in neurodegenerative processes, especially due to its changes being associated with TBI [36], temporal lobe epilepsy [37], Alzheimer’s disease [38,39], and neuropathy [13,18]. Moreover, auto-antibodies to CCL3 have been proposed as biomarkers for an advancement in human type 1 diabetes [40]. Therefore, CCL3 signaling is probably a new, important target for the development of therapeutic strategies.

In the CCI-induced neuropathic pain model, we also described an enhanced level of CCL9 (both mRNA and protein). In accordance with the immunohistochemical results from 2018 [13], CCL9 colocalizes with the NeuN marker, not with GFAP or IBA1, indicating neurons as being the main source. The neuronal origin suggests an important role of CCL9, especially in the initial stage of neuropathy. What is more, i.t. administration of a CCL9 nAb significantly diminished tactile and thermal hypersensitivity after nerve injury, which corresponds well with the results obtained earlier in the diabetic neuropathy model [13]. Taking into consideration the abovementioned results, we believe that CCL9, similar to CCL3, is a key pronociceptive factor. Although CCL9 is expressed only in rodents, the chemokine has a human ortholog, CCL23, whose upregulation was observed in the cerebrospinal fluid of patients with neuropathic pain [41]. Therefore, we consider CCL23 to be a good target for future therapeutic strategies; although, this requires further research.

Since CCL3 and CCL9 play pivotal roles in mouse neuropathic pain development, we focused our attention on the G-protein-coupled receptors of the MIP-1 family named CCR1 and CCR5. Importantly, they are present on neurons [13,42,43], microglia [39,42,44], and astrocytes [42,45]. Their presence in the neuronal cells of the spinal cord enables the important role in nociceptive transmission and explains why CCL3, CCL4, and CCL9 have strong and quick pronociceptive effects after their intrathecal administration [13,16]. Recently, a number of papers have described the involvement of CCR1 and CCR5 in the pathology of many diseases characterized by severe neuro-inflammation associated with pain [44,46,47]. The protein levels of CCR1 and CCR5 did not increase in the spinal cord after nerve injury, which is not surprising since it has also been observed in the case of other CC, e.g., CCR1, CCR5, and CCR4 in STZ- [13,48] and CCR1 and CCR3 in CCI-induced [16] neuropathic pain in mice. Many studies have indicated that both CCR1 and CCR5 have key roles in neurodegeneration [39,49,50]. Moreover, recently it was shown that CCR5 is a valid target for stroke and traumatic brain injury recovery, and the authors revealed that maraviroc improves the learning and cognition of affected animals [51]. The results of our research are especially valuable since the availability of an antagonist of CCR5, maraviroc, which is already used in the clinic, points to this receptor as a promising molecular target for future clinical trials for neuropathies of different etiologies.

Our results showed for the first time that the single intrathecal administration of a CCR1 antagonist (J113863) and CCR5 antagonists (TAK-220 and AZD-5672) dose-dependently diminished pain-related behavior after CCI. Similarly, J113863 reduces hypersensitivity in complete Freund’s [52] and diabetic [13] mouse models and maraviroc in CCI models [14,46,47]. The other CCR5 antagonist (DAPTA) was shown to be effective in the case of partial sciatic-nerve-ligation-induced hypersensitivity [53], but it was ineffective in STZ-induced [13] and CCI-induced (own unpublished data) neuropathy. This was the reason why we used the other antagonists, which are known to be strong and selective blockers of CCR5 (TAK-220 and AZD-5672). It is worth emphasizing that such good analgesic effects of the antagonists of both receptors are probably caused by the fact that numerous pleiotropic chemokines act through them. The CCR1 has ten ligands, including five with strong pronociceptive properties and well-documented spinal changes in a CCI mice model, such as CCL2, CCL3, CCL5, CCL7, and CCL9 [16,54]. The CCR5 has six ligands, including four with strong pronociceptive properties and well-documented spinal changes in a CCI model, such as CCL3, CCL5, CCL7, and CCL8 [16,54]. Importantly, these two receptors have six common ligands, which by acting through these receptors are probably seriously engaged in the formation of neuropathic pain symptoms. Therefore, our results and the available literature gave us the motivation to verify whether a simultaneous CCR1/CCR5 blockade will be more effective than blocking CCR1 or CCR5 alone. Few studies have been performed with the use of double antagonists of chemokine receptors against neuropathic pain symptoms. Kwiatkowski et al. [14] revealed that dual (cenicriviroc—CCR2/CCR5) and selective (RS504393—CCR2, maraviroc—CCR5) antagonists prevent hypersensitivity to similar degrees after repeated intrathecal injections in CCI-exposed rats. However, cenicriviroc, which blocks both receptors simultaneously, exhibited a combination of the properties of the selective antagonists (RS504393, maraviroc), which meant that, in this case, the lowered expression of the most examined pronociceptive chemokines, CCR2 and CCR5, was at the mRNA level in the spinal cord and DRGs [14]. Moreover, after single intrathecal and intraperitoneal injections in mice, cenicriviroc had the strongest analgesic properties in comparison to RS504393/maraviroc [14]. Recently, Pawlik et al. [16] showed the analgesic effectiveness of a dual CCR1/CCR3 antagonist (UCB35625). Therefore, we also wanted to check whether a simultaneous CCR1 and CCR5 blockade would be more effective than blocking each of them separately. Since no dual CCR1/CCR5 antagonist was available, we decided to use a drug combination. Our findings show that the coadministration of CCR1 and CCR5 antagonists has analgesic properties on mechanical and thermal hypersensitivity but, in general, it did not work better than selective injections. Comparable results were obtained after the coadministration of J113863 with SB328437 (CCR3 antagonist), where the common blockade did not work better on pain-related behaviors than a selective blockade [16]. It is worth remembering that after the coadministration of substances, we cannot completely rule out an interaction between them, which can impact the pharmacological effect. Moreover, we do not know how these substances mutually affect the activation of receptors and consequently the cellular response [55].

Recent animal studies have suggested that chemokines, which are important pronociceptive mediators, may also evoke a loss of the analgesic effects of opioids [56,57]; however, the exact mechanisms are still poorly understood. The results of our study clearly indicate that the neutralization of CCL3 and CCL9 improved morphine analgesia in a CCI model, which was also shown after the blockade of CCL1, CCL2, and CCL7 [9,54]. From the other groups of chemokines, we have recently shown that the neutralizing antibody against XCL1 significantly potentiates the morphine analgesia [12]. Additionally, it was also proven that blocking CXCL10 enhanced morphine antinociception in cancer-induced bone pain [58] and, from another chemokine family, CX3CL1 plays an important role in regulating morphine analgesia in naive animals [59]. Moreover, it was shown that CCL2 contributes to the development of morphine antinociceptive tolerance in rats [60]. The concept of using chemokine receptor antagonists in combination with morphine was derived from experiments conducted by our team in rodent models of neuropathic pain, e.g., RS504393 (CCR2) [61], SB328437 (CCR3) [24], C021 (CCR4) [15,48], maraviroc (CCR5) [14], and NBI-74330 (CXCR3) [62], and confirmed by others, e.g., maraviroc in inflammatory pain models [63,64]. Considering the fact that monotherapy has low effectiveness against neuropathy [65], we decided to check whether the new selective pharmacological tools can increase the effectiveness of morphine after combined administration. It is well known that apart from increasing therapeutic effectiveness, drug coadministration also reduces the risk of side effects because of the possibility of lower dose usage [66]. In our opinion, combined pharmacotherapy based on two analgesics is reasonable if the drugs used have different mechanisms of action, as in the case of modulators of opioid and chemokine system coadministration. Morphine acts selectively through all opioid receptors and is a strong mu opioid receptor (MOR) agonist and weak agonist of delta (DOR) and kappa opioid receptors (KORs) [67]. As an analgesic drug, morphine is often used in the perioperative period and in cancer therapy [68]. However, in neuropathy, this drug loses its effectiveness [2]. This results in the need for gradually increased doses, which is intrinsically linked with the increased risk of side effects [28,69,70]. The mechanisms include desensitization and internalization of opioid receptors [71,72,73,74]. There is a dependency between chemokine and opioid receptors, caused by a cross-desensitization phenomenon made possible by similarities in structure between these receptors [63]. A growing body of evidence indicates that the combination of chemokine receptor antagonists with morphine potentiates morphine’s analgesic effect in animal inflammatory [63] and neuropathic pain models [14,15,24,48,61,62]. The numerous results suggest that the interaction between opioid and chemokine receptors can be the reason for better analgesic effects. The in vitro data provide evidence that there is functional crosstalk between MOR and CCR5, whereby both of which belong to the G-protein-coupled receptor superfamily [75]. It was reported that MOR and CCR5 crosstalk is mediated by the possible creation of heterodimers of them [72,73,76,77,78]. Moreover, accumulating in vitro studies suggest that heterologous desensitization, described already for CCR5-MOR, might be responsible for better opioid efficacy [79]. In our research, we have shown that a CCR1 antagonist improves the analgesic properties of morphine, which is in agreement with results obtained in mice with diabetic neuropathy [13] and in rats in a CCI model [18]. Based on the available data, we hypothesize that the stronger analgesia of morphine in coadministration with J113863 is associated with the fact that CCR1 present on neuronal cells is coexpressed with MOR. However, it is known that the activation of CCR1 leads to the internalization of MORs, which clearly changes their function [80]. Nevertheless, this phenomenon is even more complicated because it is known that CCR1 is able to heterodimerize with CCR5 [81,82]. This fact may explain why both CCR1 (J113863) and CCR5 (TAK-220 and AZD-5672) antagonists improve the analgesic effects of morphine. In our current research, we used very selective CCR5 antagonists, and importantly, the results after their single administration in mice are consistent with those obtained after i.t. repeated administration of maraviroc in CCI-exposed rats [46]. In our study, both CCR5 antagonists coadministered with morphine were more potent than morphine alone, but only in reversing mechanical hypersensitivity. Importantly, primary pain sensations are conducted by dissimilar nerve fibers [83] and chemokines may stimulate them differently, leading to pain initiation [84,85]. This may suggest that the coadministration of CCR5 antagonists with morphine may have a stronger impact on Aβ fibers, which are responsible for a mechanical sensation [86], with less effect on Aδ and C fibers, which are responsible for feeling low temperatures [87]. This issue leads to the hypothesis that functional crosstalk between MOR and CCR5 mainly affects Aβ fibers.

Our and others’ results undoubtedly suggest that chemokine system ligands and receptors are involved in opioid analgesia in nociceptive transmission [18,24,54,63]. Given that opioid receptors can probably form heterodimers with CCR1 and CCR5, the combined administration of opioid agonists with chemokine antagonists appears to be a new, interesting strategy for the relief of chronic pain. This is particularly important because it has been shown that such drug combinations allow for the use of lower doses of opioids and consequently result in less respiratory depression [66].

## 5. Conclusions

Neuropathic pain therapy is a critical need in medicine, meaning that investigations focused on novel therapeutic targets are essential. The results show that CCL3 and CCL9, based on their spinal upregulation and the potent antinociceptive effects of their neutralizing antibodies, are probably strongly engaged in the development of neuropathic pain symptoms. Their direct neutralization not only facilitates symptoms of neuropathy but also positively affects the efficacy of morphine, which may be pivotal for making future advancements in therapy. Our results propose CCR1 and CCR5 as being targets for novel polytherapy for neuropathy. Finally, our and others’ results indicate that it is important to further investigate the role of these two chemokine receptors, since they can be key drug targets for the treatment of neuro-immunological disorders of different etiologies.

## Data Availability

The data presented in this study are available upon request from the corresponding author.

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
