# Peer review of "Pharmacological Modulation of the MIP-1 Family and Their Receptors Reduces Neuropathic Pain Symptoms and Influences Morphine Analgesia: Evidence from a Mouse Model"

_brainsci, 2023, doi:10.3390/brainsci13040579_

Round 1

Reviewer 1 Report

The study is significant and the manuscript is clearly written and presented. The author has investigated the pronociceptive properties of two MIP-1 family chemokines (CCL3 and CCL9) in neuropathic pain. 

The author should address the following comments to proceed further:

1. A similar experiment could be conducted by keeping any NSAID used to treat neuropathy as a control and comparing the results with morphine. 

2. Author should elaborate the Pharmacological modulation of chemokines and influences on morphine analgesia.

Author Response

.

Reviewer 2 Report

In the manuscript, entitled "Pharmacological modulation of the MIP-1 family and their receptors reduces neuropathic pain symptoms and influences morphine analgesia: evidence from a mouse model" Authors present an interesting study, investigating the analgesic effects of CCR1 and CCR5 blockers, in addition to neutralizing antibodies of CCL3 and CCL9.

The research question is highly relevant, and the study consists of several interesting results coming from a well-established research group with strong experience in the research area. However, there are some major points for consideration, before potential publication. Most importantly, the lack of proper control groups (sham-operated mice), which makes some of the drawn conclusions much less solid. This should be definitely addressed. My comments are listed below

Major comments

- It would be crucial to include absolute control (sham-operated) groups as well (Fig. 3, 4, 6, 7, 8, 9). The lack of absolute control makes it very hard to safely draw conclusions, and this is especially true for fig. 9. If that is not possible, at least present the pre-operation values as "healthy control". In that case, also provide reasoning in the text for not including sham mice.

- Authors should provide more details (citations or experiments) clearly showing that all the chosen drugs in the administered doses are indeed selective.

-Authors should reflect on the fact that both investigated receptors have several other ligands. CCR5 besides CCL3 also binds to CCL4 and CCL5, which are also complete agonists of the receptor (but also to CCL8, and CCL2). CCR1 seems to bind to even more ligands (CCL2, CCL3, CCL4, CCL5, CCL7, CCL8, CCL14, CCL15, CCL16, CCL23). Hence, the receptor blockade-related effects are not necessarily only a result of the blockade of CCL9 and CCL3 binding. I think the focus of the discussion might be shifted a bit and it could be also made somewhat shorter (see other comments below).

- The Methods section regarding the von Frey measurements should be further detailed. Were both hind paws measured? How many times? Do the “three touches” mean three separate measurements/paws? Was the average of threshold values presented? Were animals excluded?

- What is the pretest from the figures representing the schedule (e.g. fig 3, A)? It is not mentioned in the Methods section. Was there maybe some kind of inclusion criteria applied?

- In figs 1 and 2 the sample numbers are in a relatively wide range (n= 5-10; n= 4-10). Also, does it indicate biological replicates or sample numbers? The number of biological replicates should be also clearly indicated.

- In Fig 1 and 2: the gained results with PCR vs ELISA are not very well aligned with each other, and somewhat even contradictory at some points (e.g. day 35, CCL4 mRNA vs protein levels). Authors should reflect on these in more detail in the Discussion.

- In the Discussion, Authors indicate that the development of CCR1/CCR5 or “quadruple CCR1/CCR2/CCR3/CCR5 antagonist, may bring better analgesic effects”. However, given the various physiological roles of these receptors, this is arguably a highly questionable statement (adverse events might be also variable and serious).

- The Discussion, although interesting, reaches a bit too far away from the conducted experiments and gets a bit lengthy, but without addressing some of the results in proper detail (kindly see other comments above). E.g. “Our and others’ results undoubtedly suggest that chemokine system ligands and receptors are involved in opioid signaling in nociceptive transmission”. The experimental results of the current study only indicate the potentiation of analgesic effect. Despite being the discussion a bit lengthy, reflection on all of the gained results could be more clear, and step by step.

Minor comments:

- Authors state in the Methods that in the cold plate assay, CCI animals always reacted with the operated legs first. Was the reaction of the other leg also measured? Wouldn’t have been it more accurate than to measure only the operated leg throughout to better describe changes pre- vs postop?

- During tissue collection for PCR: were animals transfused? If not, wouldn’t blood contamination of tissues be a problem?

- For PCR and western blot it would be beneficial to also present the raw data as supplementary.

- The figure legends say: fold increase of control. Isn’t it fold increase compared to/vs control?

- Methods-results-discussion should follow the same logic. Currently, the methods start with behavioral experiments, whereas the results with PCR data.

- Why does Figure 1 not contain the results obtained by Von Frey? Also, the depiction of the experimental schedule would be more logical as panel A.

- Line 309: “more significant” is scientifically inaccurate phrasing. There are other occasions when such phrasing is used, e.g. line 377: “most significant”

-I would recommend using colors that differ more visibly. In addition, sometimes the color choosing is very unlucky because some of the lighter colors are barely visible. Also, the legends are a bit confusing, if lines do not represent significant differences throughout under the line, then rather use half tick-down (or “zig-zag”) lines.

- The Discussion starts by mentioning “activation of macrophages, microglia and astroglia”, however, such mechanistical experiments were not performed. As also discussed by the Authors later, chemokines might have other sources as well.

- Authors should refer back to the figures in the Discussion.

Author Response

.
